# Anion channel SLAH3 is a regulatory target of chitin receptor-associated kinase PBL27 in microbial stomatal closure

Yi Liu[1†], Tobias Maierhofer[2†], Katarzyna Rybak[3], Jan Sklenar[1], Andy Breakspear[4], Matthew G Johnston[4], Judith Fliegmann[5], Shouguang Huang[2], M Rob G Roelfsema[2], Georg Felix[5], Christine Faulkner[4], Frank LH Menke[1], Dietmar Geiger[2], Rainer Hedrich[2*], Silke Robatzek[1,3*]

[1]The Sainsbury Laboratory, Norwich, United Kingdom; [2]Institute for Molecular Plant Physiology and Biophysics, Julius-von-Sachs-Institute, Biocenter, University of Wuerzburg, Wuerzburg, Germany; [3]LMU Biocenter, Ludwig-Maximilian-University of Munich, Martinsried, Germany; [4]John Innes Centre, Norwich, United Kingdom; [5]Department of Plant Biochemistry, Center for Plant Molecular Biology (ZMBP), University of Tuebingen, Tuebingen, Germany

**\*For correspondence:**
hedrich@botanik.uni-wuerzburg.de (RH);
robatzek@bio.lmu.de (SR)

[†]These authors contributed equally to this work

**Competing interests:** The authors declare that no competing interests exist.

**Abstract** In plants, antimicrobial immune responses involve the cellular release of anions and are responsible for the closure of stomatal pores. Detection of microbe-associated molecular patterns (MAMPs) by pattern recognition receptors (PRRs) induces currents mediated via slow-type (S-type) anion channels by a yet not understood mechanism. Here, we show that stomatal closure to fungal chitin is conferred by the major PRRs for chitin recognition, LYK5 and CERK1, the receptor-like cytoplasmic kinase PBL27, and the SLAH3 anion channel. PBL27 has the capacity to phosphorylate SLAH3, of which S127 and S189 are required to activate SLAH3. Full activation of the channel entails CERK1, depending on PBL27. Importantly, both S127 and S189 residues of SLAH3 are required for chitin-induced stomatal closure and anti-fungal immunity at the whole leaf level. Our results demonstrate a short signal transduction module from MAMP recognition to anion channel activation, and independent of ABA-induced SLAH3 activation.
DOI: https://doi.org/10.7554/eLife.44474.001

## Introduction

Activation of the innate immune system plays an important role in the protection of body surface tissues against microbial invaders. Many agronomical important pathogens penetrate plant tissues directly through the epidermis or via stomata (*Grimmer et al., 2012*; *Faulkner and Robatzek, 2012*). These include fungal pathogens that are the causal agents of devastating diseases such as rust fungi (*Fones et al., 2017*; *Nguyen et al., 2016*; *Bebber and Gurr, 2015*; *Shafiei et al., 2007*). To defend fungal infection at the level of penetration, epidermal cells are highly immunomodulatory to oligosaccharides of fungal chitin, a microbe-associated molecular pattern (MAMP) present across fungal families (*Faulkner and Robatzek, 2012*; *McLachlan et al., 2014*; *Koers et al., 2011*). In Arabidopsis, perception of chitin is mediated at the cell surface by three related pattern recognition receptors (PRRs) (*Boutrot and Zipfel, 2017*). CHITIN ELICITOR RECEPTOR KINASE 1 (CERK1) and LYSINE MOTIF (LysM) RECEPTOR KINASE 5 (LYK5) encode single-pass transmembrane receptor kinases (*Liu et al., 2012*). LysM-domain-containing GLYCOSYLPHOSPHATIDYLINOSITOL (GPI)-anchored protein 2 (LYM2) encodes a receptor-like protein predominantly present at plasmodesmal membranes and specifically involved in cell-to-cell signalling (*Faulkner et al., 2013*). Their ectodomains consist of three LysM domains, of which in CERK1 the middle LysM domain was shown to

bind directly to chitin oligomers (*Liu et al., 2012*). Recent data demonstrate that LYK5 is the major chitin receptor in Arabidopsis, binding chitin oligomers with higher affinity than CERK1 (*Cao et al., 2014*). Upon chitin binding, LYK5 forms a complex with CERK1 leading to signal generation, which depends on CERK1 autophosphorylation at specific threonine and tyrosine residues and dephosphorylation by the CERK1-INTERACTING PROTEIN PHOSPHATASE 1 (CIPP1) phosphatase (*Cao et al., 2014*; *Liu et al., 2018*; *Suzuki et al., 2016*).

A key event of PRR signalling is the phosphorylation of receptor complex-associated RECEPTOR-LIKE CYTOPLASMIC KINASES (RLCKs), which induces the release of the RLCKs from the PRR complex and activation of downstream substrates (*Rao et al., 2018*; *Bi et al., 2018*; *Yamada et al., 2016*; *Lu et al., 2010*; *Zhang et al., 2010*). Of the 46 members of the RLCK VII subfamily, several RLCKs have been associated with pattern-triggered immunity (PTI) (*Rao et al., 2018*). BOTRYTIS-INDUCED KINASE 1 (BIK1) and the closely related PROBABLE SERINE/THREONINE-PROTEIN KINASE-LIKE 1 (PBL1) of the VII-8 subgroup induce defence signalling downstream of CERK1, the bacterial MAMP receptors FLAGELLIN SENSING 2 (FLS2) and EF-TU RECEPTOR (EFR), and the danger receptors PEP1 RECEPTOR KINASEs (PEPR) 1 and PEPR2 (*Lu et al., 2010*; *Zhang et al., 2010*; *Liu et al., 2013*). CERK1-dependent chitin signalling is shared with members of the RLCK VII-4 subgroup and PBL27 belonging to the VII-1 subgroup (*Rao et al., 2018*; *Yamada et al., 2016*), highlighting that subfamily VII RLCKs have both redundant and specific roles (*Rao et al., 2018*). Consistently, higher order mutants of VII-1, VII-4 and VII-8 subgroup members are all compromised in the chitin-induced burst of reactive oxygen species (ROS) while showing different response signatures depending on the type of MAMP and defence reaction (*Rao et al., 2018*). Being a direct substrate of CERK1, initial studies suggest that *pbl27* mutants were not defective in chitin-induced ROS burst but MITOGEN -ACTIVATED PROTEIN KINASES 3/6 (MPK3/6) signalling and callose deposition (*Yamada et al., 2016*; *Shinya et al., 2014*). PBL27 was shown to phosphorylate MAPKKK5 in a CERK1-dependent manner, resulting in the dissociation of MAPKKK5 from PBL27 and activation of MKK4 and MKK5, upstream kinases of the MAPK signalling cascade (*Yamada et al., 2016*). However, these findings are recently challenged, since chitin-triggered MAPK activation was not compromised in single *pbl27* and higher order *rlck vii-1* mutants (*Rao et al., 2018*), suggesting that more research is required. Instead, subgroup VII-4 members are involved in activation of MPK3/6 by chitin, yet not immune signalling by bacterial flagellin (*Ranf et al., 2014*). On the other hand, BIK1 and PBL1 are required for flagellin-induced ROS production but not MAPK activation (*Zhang et al., 2010*; *Li et al., 2014*). This highlights differences between RLCK-mediated signalling in response to fungal and bacterial MAMPs.

MAMP perception results in the closure of stomata, pores formed by a guard cell pair, and thereby promotes plant tissue surface immunity (*McLachlan et al., 2014*; *Melotto et al., 2006*). As a counterstrategy and demonstrating the importance of stomatal closure, infectious pathogens secrete effectors, which function to inhibit closure of stomata and to induce stomatal opening, or lock stomata in the wide open state by fungal-produced fusicoccin (*McLachlan et al., 2014*; *Lozano-Durán et al., 2014*; *Melotto et al., 2006*). Reducing the complexity of whole plant/organ systems with different cell types, guard cells have been well established as a single cell model system and used to dissect both immune- and ABA-signalling (*Qi et al., 2017*). Stomatal apertures are controlled by cell volume changes triggered upon ion fluxes (*McLachlan et al., 2014*). Stomatal closure to bacterial flagellin requires activation of S-type anion channels mediated by the SLOW ANION CHANNEL-ASSOCIATED 1 (SLAC1), a weak rectifying anion channel present at the plasma membrane of guard cells, and closely related SLAC1 HOMOLOG 3 (SLAH3) (*Guzel Deger et al., 2015*; *Montillet et al., 2013*). Exposure to chitin oligosaccharides (herein referred to as chitin) and chitosan, a deacetylated derivative of chitin, reduce stomatal apertures (*Bourdais et al., 2019*; *Klüsener et al., 2002*). Evidence suggest that chitosan stimulates S-type anion channels (*Koers et al., 2011*), but the molecular components involved in channel activation and thus promoting stomatal closure to chitin and derivatives remain elusive.

In abiotic stress signalling, SLAC1 is activated by OPEN STOMATA 1 (OST1), a SUCROSE NON-FERMENTING 1 (SNF1)-related protein kinase (SnRK), which involves S120 phosphorylation of the SLAC1 N-terminus and is independent of elevated cytoplasmic calcium (*Geiger et al., 2010*; *Vahisalu et al., 2010*; *Geiger et al., 2009*). Yet, elevation of cytosolic calcium also activates S-type anion channels (*Stange et al., 2010*; *Schroeder and Hagiwara, 1989*), consistent with the findings that CALCIUM-DEPENDENT PROTEIN KINASE 3 (CPK3) and CPK21 activate SLAC1 (*Geiger et al.,*

*2010*; *Scherzer et al., 2012*). CPK6 and CPK23 also appear to activate SLAC1 but largely independent of cytosolic calcium elevation (*Geiger et al., 2010*; *Scherzer et al., 2012*), which involves S59 phosphorylation at the SLAC1 N-terminus (*Brandt et al., 2012*). Moreover, calcium sensitive CALCINEURIN B-LIKE 1 (CBL1) and CBL9 together with CBL-INTERACTING PROTEIN KINASE 23 (CIPK23) are capable of SLAC1 activation, through a phosphorylated residue distinct from OST1 phosphorylation (*Maierhofer et al., 2014a*). This demonstrates that different kinases regulate SLAC1 activity at distinct phosphorylation sites. SLAH3 is only activated upon co-expression of CPKs and CBL/CIPKs but not by OST1 (*Maierhofer et al., 2014a*; *Geiger et al., 2011*), suggesting a different mode of regulation.

Previous studies suggested several kinases in guard cell signalling downstream of FLS2, including BIK1, MPK3/6 and OST1 (*Kadota et al., 2014*; *Li et al., 2014*; *Guzel Deger et al., 2015*; *Montillet et al., 2013*). Thus, although the pathways conferring MAMP-induced stomatal closure are emerging, the molecular events that result in anion release upon PRR signalling are not understood. Here, we identify that the Arabidopsis LYK5-CERK1-PBL27 receptor complex is responsible for chitin-induced stomatal closure and anti-fungal immunity. Using biochemical and molecular approaches, we show that PBL27 directly interacts with and phosphorylates SLAH3, consistent with SLAH3 phosphorylation induced by chitin. PBL27 phosphorylates S127 and S189 residues of SLAH3, which are required for chitin-induced stomatal closure and anti-fungal immunity. Both phospho-sites are critical for PBL27-mediated activation of SLAH3 and revealing a role for S189 in CERK1-dependent amplification of SLAH3 opening. Taken together, our data identified an S-type anion channel as a novel target of RLCKs with relevance to immune defences.

## Results

### LYK5-CERK1, PBL27 and SLAH3 are involved in stomatal closure

To better understand LYK-type components of chitin-induced stomatal closure in Arabidopsis, we measured the stomatal response in mutants of all LYK family members (*Willmann et al., 2011*). Single null *cerk1* and *lyk5* mutant plants showed no closure of stomata in response to chitin treatment, whilst *lyk2*, *lyk3*, and *lyk4* behaved like wild-type plants (*Figure 1A*; *Figure 1—figure supplement 1*). This is in agreement with LYK5 representing the major chitin receptor and inducing chitin signalling through complex formation with CERK1 (*Cao et al., 2014*). Given that PBL27 is preferentially phosphorylated by CERK1 signalling (*Shinya et al., 2014*), consistently stomata of *pbl27* mutants showed no closure to chitin either (*Figure 1B*; *Figure 1—figure supplement 1*). Transgenic expression of respective C-terminally tagged green fluorescent protein (GFP) fusion variants restored chitin-induced stomatal closure in *cerk1*, *lyk5* and *pbl27* mutants (*Figure 1—figure supplement 1*).

Assuming that S-type anion channels are downstream targets of CERK1-induced guard cell signalling, we tested whether loss of SLAC1 or SLAH3 would result in impaired chitin-induced stomatal closure. Unexpectedly, single *slah3* mutants, which are sensitive to pattern-induced stomatal closure (*Zheng et al., 2018*; *Guzel Deger et al., 2015*), failed to close stomata in response to the fungal MAMP (*Figure 1C*). Single *slac1* mutants exhibited wild type-like chitin-induced stomatal closure (*Figure 1C*). Chitosan can stimulate S-type channel activity in barley and reduces stomatal apertures in Arabidopsis (*Koers et al., 2011*; *Klüsener et al., 2002*). Yet, in this case, stomata of *slah3* mutant showed intermediate levels of closure, being not significantly different to stomatal apertures in chitosan-stimulated wild type guard cells and unstimulated guard cells of *slah3* (*Figure 1—figure supplement 2*). It is possible that stomatal closure to chitosan involves redundant functions with SLAC1, as shown for FLS2 and PEPR1/2 guard cell signalling (*Guzel Deger et al., 2015*; *Zheng et al., 2018*). Most crude chitin preparations, i.e. as used in this study, represent a mixture of long-chain and short-chain chitin and chitosan oligosaccharides, which could be perceived by different receptors. CERK1 binds both chitin and chitosan with a preference for long-chain oligosaccharides (*Petutschnig et al., 2010*), suggesting that it is a major receptor in chitin-induced closure of stomata. That pattern-induced stomatal closure is fully (chitin) and partially (chitosan, flagellin [*Guzel Deger et al., 2015*], danger signals [*Zheng et al., 2018*]) dependent on SLAH3 points at an important role of the channel in this process. Together, our genetic data suggest that chitin-induced stomatal closure is mediated by LYK5-CERK1-PBL27 receptor complex components, and primarily

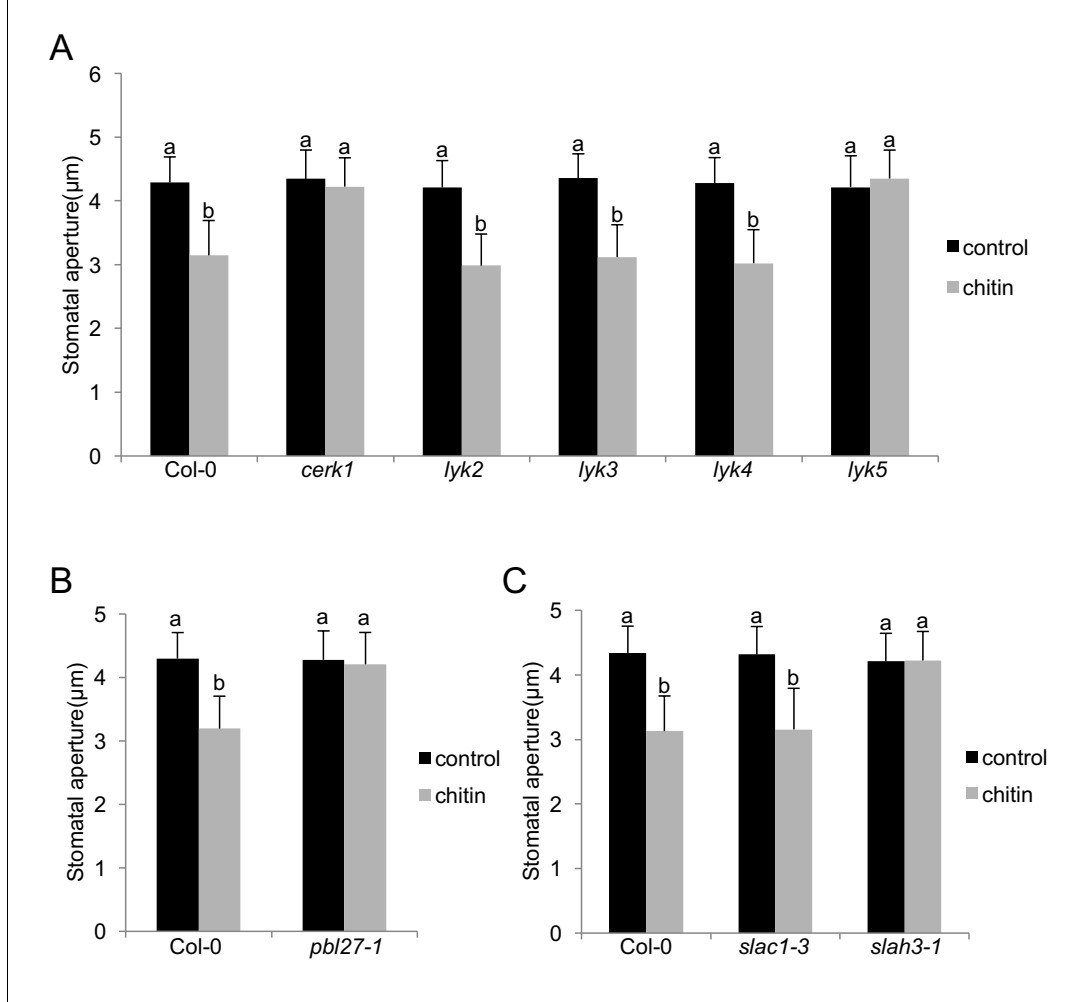

**Figure 1.** The LYK5-CERK1-PBL27 receptor complex and SLAH3 are required for chitin-induced stomatal closure. (**A–C**) Stomatal aperture measurements in mutants of all members of the LysM-RLK family (**A**), *pbl27* (**B**), *slac1* and *slah3* (**C**). Mature leaf discs were soaked in opening buffer (10 mM MES, 50 mM KCl, pH 6.15) and kept under light (100 μmol.m-2 s −1) for 2 hr. Stomatal apertures were measured 2 hr after treatment with 1 mg/ml chitin. Values are mean ± SD (n > 60; two-way ANOVA). Different letters indicate significantly different values at p<0.05. These experiments were repeated three times with similar results.

DOI: https://doi.org/10.7554/eLife.44474.002

The following source data and figure supplements are available for figure 1:

**Source data 1.** Source data for stomatal measurements shown in *Figure 1* and *Figure 1—figure supplements 1* and *2*, and for *Figure 4*.
DOI: https://doi.org/10.7554/eLife.44474.005
**Figure supplement 1.** Functional complementation and guard cell expression of the LYK5-CERK1-PBL27 receptor complex.
DOI: https://doi.org/10.7554/eLife.44474.003
**Figure supplement 2.** Current ejection of chitosan induces stomatal closure in Arabidopsis.
DOI: https://doi.org/10.7554/eLife.44474.004

involving SLAH3. Therefore, we focused our study on the molecular and biochemical characterization of SLAH3 stimulation by CERK1 signalling.

## SLAH3 interacts with and is phosphorylated by PBL27

We next sought to test which of the identified components are directly linked to each other and performed bimolecular fluorescence complementation (BiFC) experiments. After transient expression of SLAH3 together with CERK1 and PBL27 in *N. benthamiana*, we observed a clear fluorescent signal of SLAH3-YFPn and PBL27-YFPc but not SLAH3-YFPn and CERK1[D441V]-YFPc (*Figure 2A*). Of note, we used kinase dead (KD) transient expression of CERK1[D441V]-YFPc, which is competent to interact

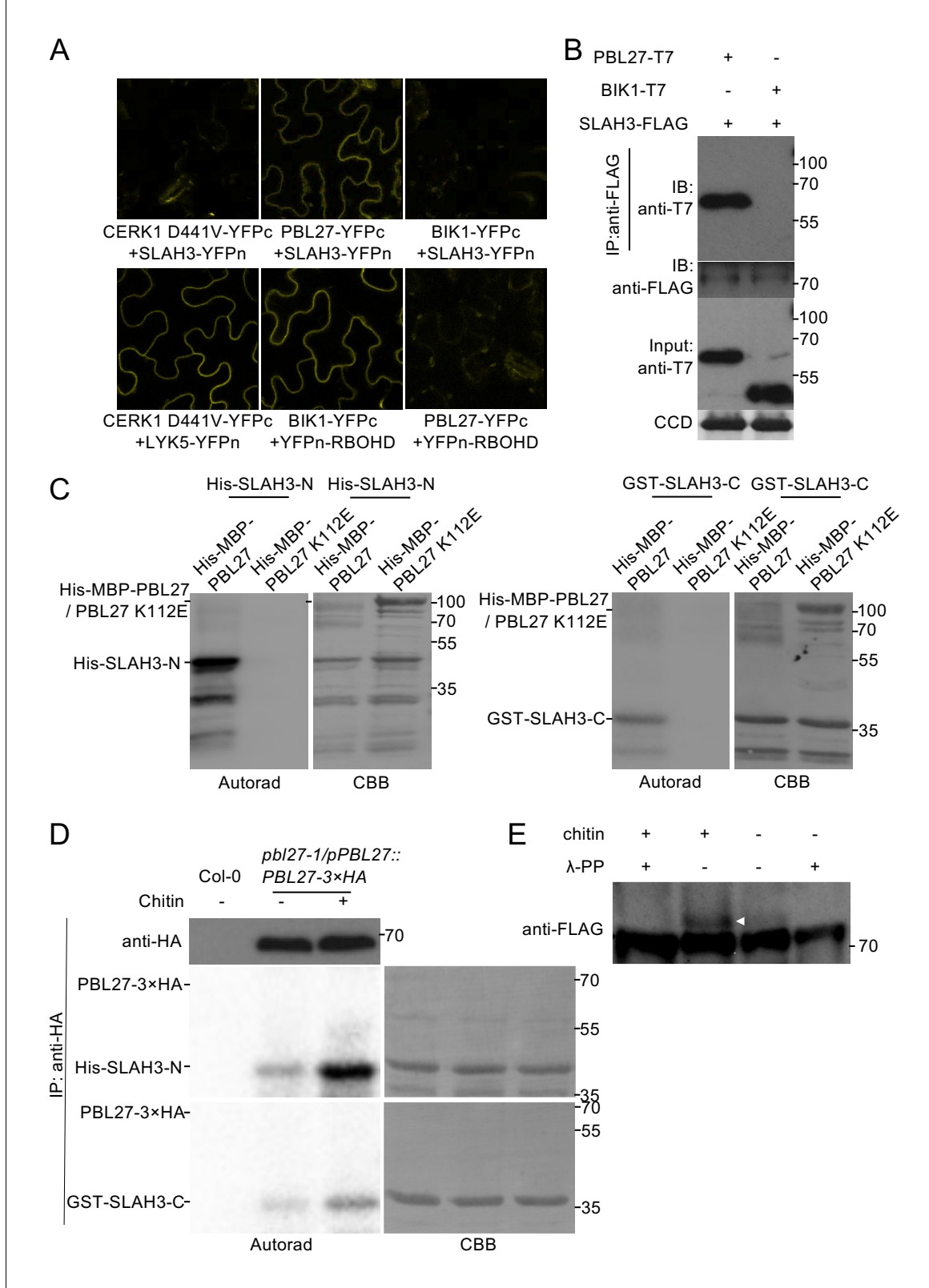

**Figure 2.** PBL27 interacts with and phosphorylates SLAH3. (**A**) Confocal microscopy of *N. benthamiana* leaves transiently expressing the indicated split-YFP constructs. Representative images are shown. (**B**) Co-immunoprecipitation of PBL27 and SLAH3 transiently expressed in *N. benthamiana* leaves. These experiments were performed at least twice with similar results. Expected sizes of PBL27-T7 and BIK1-T7 fusion proteins correspond to 57 kDa and 46 kDa, respectively. SLAH3-FLAG used for immune-precipitation has an expected size of 73 kDa. (**C**) PBL27 *trans*-phosphorylates SLAH3-N and

*Figure 2 continued on next page*

*Figure 2 continued*

SLAH3-C. In vitro kinase assay incubating equal amounts of recombinant His-MBP-PBL27, His-MBP-PBL27 K112E (kinase dead) with recombinant His-SLAH3-N or GST-SLAH3-C (GST-SLAH3-C-His). Autoradiogram, left panel; Coomassie colloidal blue (CCB) stained membrane, right panel. These experiments were repeated three times with similar results. (D) Chitin-activated PBL27 *trans*-phosphorylates SLAH3-N. Transgenic *pbl27-1/pPBL27::PBL27−3 × HA* Arabidopsis seedlings were treated (+) or not (-) with 1 mg/ml chitin for 10 min. Total proteins were subjected to immunoprecipitation with anti-HA beads followed by immunoblot analysis with anti-HA to reveal PBL27−3 × HA (upper panel). Immuno-precipitated PBL27−3 × HA was incubated with recombinant His-SLAH3-N for in vitro kinase assay. Autoradiogram, left panel; Coomassie colloidal blue (CCB) stained membrane, right panel. Col-0 seedlings were used as a control. These experiments were repeated three times with similar results. (E) Chitin induces SLAH3 phosphorylation. Transgenic Arabidopsis *slah3-1/35S::SLAH3−3 × FLAG* transgenic were treated (+) or not (-) with 1 mg/ml chitin for 30 min. Total proteins were subjected to immunoprecipitation with anti-FLAG beads. The phosphorylated form of SLAH3−3 × FLAG was shifted upward in Phos-tag SDS-PAGE. After phosphatase treatment, the shifted band of SLAH3−3 × FLAG dispersed, indicating the SLAH3−3 × FLAG was phosphorylated after treatment with chitin. The white arrow indicates the phosphorylated form of SLAH3-FLAG. The bands were detected with an anti-FLAG antibody. This experiment was repeated three times with similar results.

DOI: https://doi.org/10.7554/eLife.44474.006

The following source data and figure supplement are available for figure 2:

**Source data 1.** Source data for BiFC images shown in *Figure 2*.
DOI: https://doi.org/10.7554/eLife.44474.008
**Source data 2.** Source data for co-IP blots images shown in *Figure 2*.
DOI: https://doi.org/10.7554/eLife.44474.009
**Source data 3.** Source data for blots on in vitro phosphorylation shown in *Figure 2*.
DOI: https://doi.org/10.7554/eLife.44474.010
**Source data 4.** Source data for blots on in vitro-in vivo phosphorylation shown in *Figure 2*.
DOI: https://doi.org/10.7554/eLife.44474.011
**Source data 5.** Source data for blots on in vivo phosphorylation shown in *Figure 2*.
DOI: https://doi.org/10.7554/eLife.44474.012
**Figure supplement 1.** Identification of SLAH3 tryptic phospho-peptides.
DOI: https://doi.org/10.7554/eLife.44474.007

with LYK5-YFPn (*Figure 2A*), as wild type CERK1 promoted cell death in *N. benthamiana*. Importantly, co-expression of SLAH3-YFPn with BIK1-YFPc, also competent to associate with CERK1 (*Zhang et al., 2010*), did not reconstitute a detectable fluorescent signal (*Figure 2A*). By contrast, co-expression of BIK1-YFPc but not PBL27-YFPc with YFPn-RBOHD generated fluorescent signals (*Figure 2A*), consistent with the notion that BIK1 signalling activates the RBOHD-mediated ROS burst to flagellin and that single *pbl27* mutants are not compromised in chitin-induced ROS (*Kadota et al., 2014*; *Li et al., 2014*; *Shinya et al., 2014*). If reconstituted, fluorescent signals were detected at the cell periphery, which is consistent with the subcellular localization pattern of functional LYK5-GFP, CERK1-GFP and PBL27-GFP fusion proteins (*Shinya et al., 2014*; *Erwig et al., 2017*), and as previously described for SLAH3 (*Demir et al., 2013*). To independently confirm the PBL27-SLAH3 interaction in *N. benthamiana*, we immuno-precipitated SLAH3-FLAG and observed that it associated with PBL27-T7 but not BIK1-T7, and that the association was independent of chitin stimulation (*Figure 2B*). Together, these results show that SLAH3 exists in a complex with PBL27 in a chitin-independent manner.

Although co-immunoprecipitation provides evidence that proteins associate in a complex together, it does not verify direct protein interactions. As PBL27 was shown to directly phosphorylate MAPKKK5 (*Yamada et al., 2016*), we tested in vitro if PBL27 directly phosphorylates SLAH3. In vitro phosphorylation experiments using N- and C-terminal domains of SLAH3 indicated that PBL27, but not its kinase dead variant PBL27$^{K112E}$, *trans*-phosphorylated both the N-and C-terminal domain of SLAH3 (*Figure 2C*). Of note, a weaker in vitro trans-phosphorylation could be observed for SLAH3-C compared with SLAH3-N (*Figure 2C*). The finding that PBL27 phosphorylates SLAH3 in vitro also provides evidence for a direct interaction between these two proteins, which is consistent with our BiFC and co-immunoprecipitation results (*Figure 2A and B*).

To test if SLAH3 phosphorylation by PBL27 is regulated in a chitin-dependent manner, we immuno-precipitated PBL27-HA from transgenic Arabidopsis plants (*Shinya et al., 2014*), which were untreated or treated with chitin, and then subjected PBL27-HA to *trans*-phosphorylation experiments with N-terminal and C-terminal SLAH3 in vitro. Both SLAH3-N and SLAH3-C were clearly phosphorylated when incubated with PBL27-HA, which was significantly stronger when

PBL27-HA was purified from chitin-treated plants (*Figure 2D*). This revealed that PBL27 is able to *trans*-phosphorylate SLAH3-N and SLAH3-C depending on chitin stimulation. We next immuno-precipitated SLAH3-FLAG from untreated and chitin-induced transgenic Arabidopsis plants and then analysed SLAH3 phosphorylation status using Phos-tag SDS-PAGE. Elicitation with chitin led to SLAH3 phosphorylation, as indicated by a band shift, which can be reversed upon phosphatase treatment (*Figure 2E*). Together, our in vitro and in vivo data provide evidence that SLAH3 phosphorylation status can be regulated by PBL27 and depends on chitin stimulation.

## PBL27 activates S-type anion currents mediated by SLAH3

Given that phosphorylation at the N-terminus of Arabidopsis SLAC/SLAH anion channels by different kinases results in S-type anion channel activation (*Geiger et al., 2010*; *Vahisalu et al., 2010*; *Geiger et al., 2009*; *Scherzer et al., 2012*; *Brandt et al., 2012*; *Maierhofer et al., 2014a*), we next tested whether SLAH3 phosphorylation by PBL27 also induced SLAH3-derived anion currents. To this end, we performed measurements in *Xenopus* oocytes, in which all investigated components were introduced by cRNAs injection, as the cells very efficiently translate cRNAs (*Tammaro et al., 2008*). In two-electrode voltage clamp (TEVC) experiments SLAH3 remained silent when it was injected into *Xenopus* oocytes alone (*Figure 3A*) (*Geiger et al., 2011*). By contrast, co-injection with PBL27 generated macroscopic S-type anion currents (*Figure 3A*), with current amplitudes of 8 µA similar to SLAH3 activated by calcium-insensitive CPK21∆EF or CBL1/CIPK23 (*Figure 3B*) (*Maierhofer et al., 2014a*; *Geiger et al., 2011*). PBL27-mediated activation of SLAH3 was not suppressed upon co-injecting the protein phosphatase ABA INSENSITIVE 1 (ABI1), while ABI1 inhibited the CBL1/CIPK23-dependent activation of SLAH3 (*Figure 3—figure supplement 1A*). This suggests that PBL27 is sufficient to trigger SLAH3-mediated anion transport independent of ABA signalling (*Geiger et al., 2010*). Here it should be noted that PBL27 did not activate SLAC1, unlike OST1 and CBL1/CIPK23 (*Figure 3—figure supplement 1B*) and which is consistent with wild type-like stomatal closure in *slac1* mutants (*Figure 1C*).

Considering the evidence for RLCK members of the VII-4 and VII-8 subgroups as well as MAPKKK5 in CERK1-dependent signalling (*Rao et al., 2018*; *Yamada et al., 2016*; *Zhang et al., 2010*), we explored whether SLAH3 could be activated by these kinases. However, co-injection with PBL19 or PBL39, both belonging to the VII-4 subgroup involved in chitin-induced ROS production and MAPK activation (*Rao et al., 2018*), did not induce SLAH3 anion channel activity (*Figure 3—figure supplement 1C*). A putative role for SLAH3 regulation by MAPKs is further excluded, as co-injection with MAPKKK5, a direct substrate of PBL27 (*Yamada et al., 2016*), did neither activate SLAH3 directly nor influence PBL27-mediated activation of SLAH3 (*Figure 3—figure supplement 1D*). Also, the CERK1-associated BIK1 kinase (*Zhang et al., 2010*) did not induce anion currents when co-injected with SLAH3 (*Figure 3—figure supplement 1C*). PBL1 is closely related with and functionally redundant to BIK1, and also implicated in chitin defences (*Zhang et al., 2010*). We therefore tested PBL1 and observed that it activated SLAH3, inducing current amplitudes even higher compared with PBL27 injection (*Figure 3—figure supplement 1C*). Thus, in addition to PBL27, the VII-8 subgroup member PBL1 can produce SLAH3-mediated S-type anion currents.

## PBL27-mediated SLAH3 activation is increased by signalling competent CERK1

PBL27 is a direct substrate of the LYK5-CERK1 complex (*Yamada et al., 2016*; *Shinya et al., 2014*). We therefore investigated the contribution of the receptor kinases for SLAH3 activation by PBL27. SLAH3-mediated anion current amplitudes were strongly increased in the presence of CERK1 but not LYK5 co-injection, when activated by PBL27 (*Figure 3C*). The positive regulation of SLAH3 opening by CERK1 is even more striking when the steady-state currents are blotted against the voltage (*Figure 3—figure supplement 2A, B and C*). The enhanced activity is due to a shift of the relative open probability of SLAH3 to more negative membrane potentials in the presence of CERK1. When LYK5 and CERK1 were co-injected without PBL27, SLAH3 remained silent (*Figure 3C*), corresponding with SLAH3 interaction with PBL27 but not CERK1 (*Figure 2A*). Consistent with the finding that PBL27 is phosphorylated by CERK1 (*Shinya et al., 2014*), only signalling-competent CERK1 amplified PBL27-mediated SLAH3 activation (*Figure 3—figure supplement 2D*). Activation of SLAH3 by PBL27 was dependent on its kinase activity (*Figure 3B*), which was also necessary for amplified

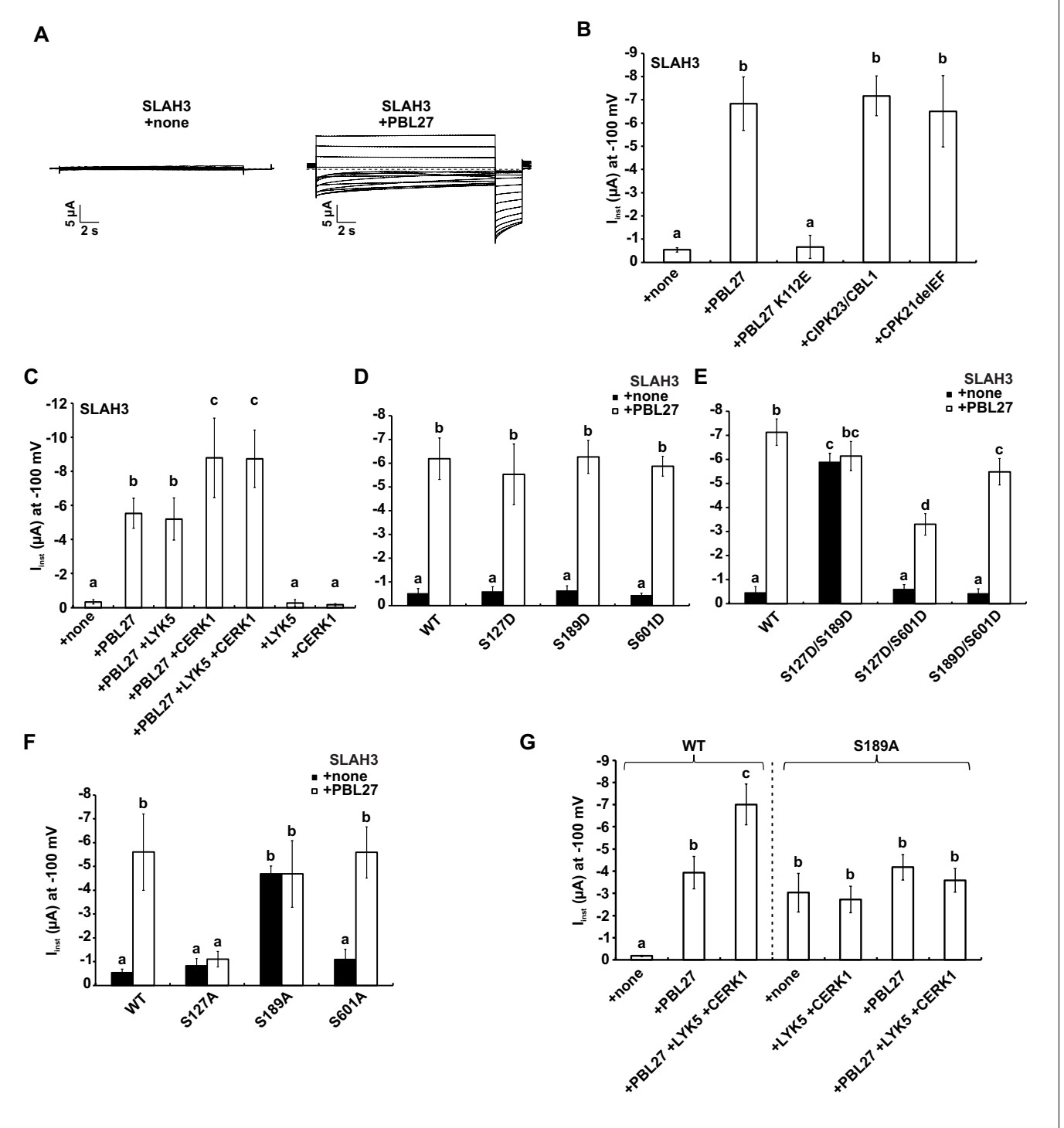

**Figure 3.** S-type anion currents are activated by co-injection of SLAH3 and PBL27 in oocytes. (A) Macroscopic currents of *Xenopus* oocytes expressing SLAH3 in the presence or absence of PBL27 in response to the standard voltage protocol. Currents were recorded in 30 mM nitrate-based buffers. Representative cells are shown. (B) Instantaneous currents ($I_{inst}$) at −100 mV recorded from oocytes injected with SLAH3 alone or co-injecting SLAH3 with the indicated kinases in the presence of 100 mM nitrate (n ≥ 4; mean ± SD). PBL27 K112E represents a kinase-dead mutant. CPK21DEF represents a $Ca^{2+}$-independent and thus constitutive active truncation mutant of CPK21. (C) Instantaneous currents ($I_{inst}$) at −100 mV of oocytes injected with WT SLAH3 alone or co-injected with PBL27, LYK5, CERK1 or different combinations of these components as indicated in the figure. Currents were recorded in nitrate-based buffers (100 mM) (n ≥ 4; mean ± SD). (D) and (E) Instantaneous currents ($I_{inst}$) of oocytes injected with SLAH3 WT or (D) the phospho-mimetic single mutants S127D, S189D and S601D or (E) the phospho-mimetic double mutants S127D/S189D, S127D/S601D or S189D/S601D were measured in the presence or absence of PBL27 at −100 mV. Currents were recorded in standard buffers containing 100 mM nitrate (n ≥ 4; mean ± SD).
*Figure 3 continued on next page*

*Figure 3 continued*

(F) Instantaneous currents ($I_{inst}$) at $-100$ mV of oocytes injected with SLAH3 WT or the phospho-dead mutants S127A, S189A and S601A in the presence or absence of PBL27 in nitrate-based buffers (100 mM) (n $\geq$ 4; mean ± SD). (G) Instantaneous currents ($I_{inst}$) at $-100$ mV in nitrate-based buffers of oocytes injected with WT SLAH3 or the mutant S189A alone or co-injected with PBL27, LYK5 and CERK1 as indicated in the figure (n $\geq$ 4; mean ± SD). (B) - (G) Significant differences (ANOVA with Tukey's HSD test, p<0.01) between bars are denoted with different letters.

DOI: https://doi.org/10.7554/eLife.44474.013

The following source data and figure supplements are available for figure 3:

**Source data 1.** Source data for current measurements shown in *Figure 3* and *Figure 3—figure supplements 1* and *2*.
DOI: https://doi.org/10.7554/eLife.44474.017
**Figure supplement 1.** Activation of SLAH3 and its close relative SLAC1 by different kinases.
DOI: https://doi.org/10.7554/eLife.44474.014
**Figure supplement 2.** CERK1 enhances PBL27 activation of SLAH3.
DOI: https://doi.org/10.7554/eLife.44474.015
**Figure supplement 3.** Activation of SLAH3 wild type and the SLAH3 mutants S127A, S189A and S601A.
DOI: https://doi.org/10.7554/eLife.44474.016

SLAH3 activation by CERK1 (*Figure 3—figure supplement 2E*). This suggests that the phosphorylation status of PBL27 correlates with the activation of SLAH3 and is in agreement with enhanced phosphorylation of SLAH3 when PBL27 was purified from chitin-stimulated cells (*Figure 2D*). We next tested whether PBL1 activation of SLAH3 is downstream of CERK1 signalling. Unexpectedly, co-injection with CERK1 inhibited PBL1 activation of SLAH3 (*Figure 3—figure supplement 1E*). It is possible that PBL1 regulates SLAH3 downstream of PEPR1/2 signalling (*Liu et al., 2013*; *Zheng et al., 2018*), and becomes inactive for *trans*-phosphorylation of SLAH3 in the presence of CERK1. We conclude that PBL27 is the primary kinase that directly phosphorylates SLAH3 for the release of anions and functions in an activation status-dependent manner regulated by CERK1.

## PBL27 activates SLAH3 anion transport through specific phospho-sites

Having found that PBL27 phosphorylates SLAH3 (*Figure 2C and D*), these results are consistent with the hypothesis that PBL27 activates SLAH3 through phosphorylation. For that reason, we sought to identify residues that are phosphorylated by PBL27 and tested whether these sites are required for SLAH3 activation. To identify the SLAH3 phosphorylation sites resulting from PBL27 *trans*-phosphorylation, SLAH3-N and SLAH3-C were incubated with PBL27 in vitro, and the phosphorylation sites were determined by liquid chromatography-tandem mass spectrometry (LC-MS/MS) analysis. We discovered that SLAH3 was phosphorylated at three sites by PBL27: S127 and S189 at SLAH3-N and S601 at SLAH3-C (*Figure 1—figure supplement 1A, B and C*).

We next examined the functional relevance of PBL27-mediated S127, S189 and S601 phosphorylation for the activation of SLAH3. In oocytes, injection of single phospho-mimic mutations of S127, S189 and S601 (S127D, S189D and S601D) did not auto-activate SLAH3 and had no impact on PBL27 activation of SLAH3 (*Figure 3C*). However, remarkably, the double mutant S127D/S189D, but neither S127D/S601D nor S189D/S601D, was constitutive active (*Figure 3E*). Current amplitudes of the mutants were comparable with wild type SLAH3 activated by PBL27 (*Figure 3D*). Interestingly, co-expression of S127D/S189D with PBL27 did not result in a further increase in anion currents whereas S127D/S601D and S189D/S601D could still be activated by PBL27 (*Figure 3E*). These results provide evidence that the phosphorylation status of both S127 and S189 is critical for SLAH3 channel opening.

We tested single phospho-dead mutations of S127, S189 and S601 (S127A, S189A and S601A) for the activation of SLAH3. Importantly, we observed that PBL27 activation of SLAH3 is abrogated by S127A, but not S189 and S601 (*Figure 3F*). We also found that S127A did not affect the activation of SLAH3 by other kinases such as CPK21ΔEF and CBL1/CIPK23 or by the heteromerization with the silent anion channel subunit SLAH1 (*Figure 3—figure supplement 3A*) (*Cubero-Font et al., 2016*). These results demonstrate that PBL27 regulates SLAH3 activation through phosphorylation of S127, a site not required for CPK21 and CBL1/CIPK23 activation of SLAH3 (*Figure 3—figure supplement 3A*). However, whilst S127 phosphorylation is required for SLAH3 activation by PBL27, it is not sufficient by itself to induce anion transport (*Figure 3D*). It can only auto-activate SLAH3 in the

context of additional S189 phospho-mimicry (*Figure 3E*), and thus suggests a functional link between these two residues in SLAH3 channel opening.

Surprisingly, the S189A instead of the S189D mutation resulted in constitutive activation of SLAH3 (*Figure 3D and F*). The S189A auto-activation could not be further elevated by co-injection of PBL27 or other anion channel activating components like CIPK23/CBL1 or SLAH1 (*Figure 3F*, *Figure 3—figure supplement 3B*). However, CERK1-mediated amplification of SLAH3 activation by PBL27 is compromised in the S189A constitutive active variant of SLAH3 (*Figure 3G*). Thus, it seems that PBL27 phosphorylation of S127 activates SLAH3 but full opening of the SLAH3 channel additionally requires PBL27 phosphorylation of S189, in a CERK1-dependent manner. However, it is possible that S127 also contributes to amplified SLAH3 opening by CERK1, which we are unable to address as a S127A/S189A variant would already affect the PBL27 pre-activation of SLAH3. S601 seems not to be involved in phosphorylation-dependent or -independent activation of SLAH3, since the mutant S601A could be either activated by PBL27 (*Figure 3F*) or by CPK21ΔEF, CIPK23/CBL1 and SLAH1 (*Figure 3—figure supplement 3C*). This is in line with single or double phospho-mimic mutants including S601D that do not affect SLAH3 activity (*Figure 3D and E*).

## SLAH3 is required for chitin-induced stomatal closure and anti-fungal immunity

We finally evaluated the relevance of the PBL27-dependent SLAH3 phospho-sites in chitin-induced stomatal closure. We generated stable transgenic Arabidopsis lines expressing wild type SLAH3 and the single S127A, S189A and S601A phospho-dead variants in the *slah3* mutant background. We tested each six independent T1 lines that all exhibited significant transcription of the transgene (*Figure 4—figure supplement 1*). Transgenic *slah3* plants expressing wild-type SLAH3 showed chitin-induced stomatal closure (*Figure 4A*). By contrast, plants expressing the S127A and S189A variants failed to close stomata in response to chitin, similar to *slah3* mutants (*Figure 4B and C*). This is consistent with our result that S127 phosphorylation is required for activation of SLAH3 (*Figure 3F*). The observation that plants expressing the S189A variant were impaired in stomatal closure is in apparent contrast with its auto-activity (*Figure 3F*). It would be expected that auto-active SLAH3 triggers a continual closure of stomata and such plants would be severely affected in development. Since we could recover transgenic plants expressing SLAH3 S189A (*Figure 4—figure supplement 1*) and having no obvious developmental phenotype suggest two scenarios: Firstly, constitutive negative regulation at a residue that could compensate the effect of S189A, i.e. S127 (*Figure 3E*). This could result in insensitivity to chitin stimulation and open stomata (*Figure 4C*). Secondly, the level of S189A auto-activation of SLAH3 is below a threshold of triggering continual stomatal closure. Since S189A is compromised in CERK1-dependent full activation of SLAH3 by PBL27 (*Figure 3G*, *Figure 3—figure supplement 2A, B and C* note, in the presence of CERK1, the SLAH3/PBL27 pair mediated three times higher currents), stomata do not close in response to chitin (*Figure 4C*). However, although CIPK23/CBL1 were unable to increase SLAH3 opening beyond the level of its auto-activation (*Figure 3—figure supplement 3B*), we cannot exclude the possibility that the auto-activation of SLAH3-S189A in oocytes hints at a structurally important role in SLAH3 function that is not only specific to the activation of PBL27 by CERK1. As expected from oocyte experiments (*Figure 3F*), plants expressing the S601A SLAH3 variant behaved wild type-like (*Figure 4D*).

Unfortunately, pathosystems that could be used to explore the biological significance of chitin-induced stomatal closure to the outcome of anti-fungal immunity are of limited availability in Arabidopsis (*Cheng et al., 2012*; *Cheng et al., 2013*). Instead, we thought to explore anti-fungal resistance in whole leaves, because SLAH3 is expressed in both guard cells and mesophyll (*Geiger et al., 2011*), and infection of *Botrytis cinerea* and *Alternaria brassicicola* was supported in *bik1* and *pbl27* mutants, respectively (*Shinya et al., 2014*; *Veronese et al., 2006*). We assayed *slah3* mutants and the single S127A, S189A and S601A phospho-dead SLAH3 lines in the *slah3* background for resistance to necrotrophic fungus *B. cinerea*. Lesion diameter analysis revealed that relative to Col-0 and complementation lines expressing SLAH3 wild type, mutant *slah3* plants and complementation lines expressing S127A and S189A, but not S601A, developed larger disease lesions three days post inoculation (*Figure 5A*). Notably, *slah3* mutants were not generally immune-compromised, since we observed wild type-like chitin-induced ROS burst and ethylene production (*Figure 5B and C*). MAMP-induced ethylene production appears to be downstream of MAPK activation (*Bethke et al., 2009*; *Liu and Zhang, 2004*). We therefore assume that SLAH3-mediated resistance to *B. cinerea* is

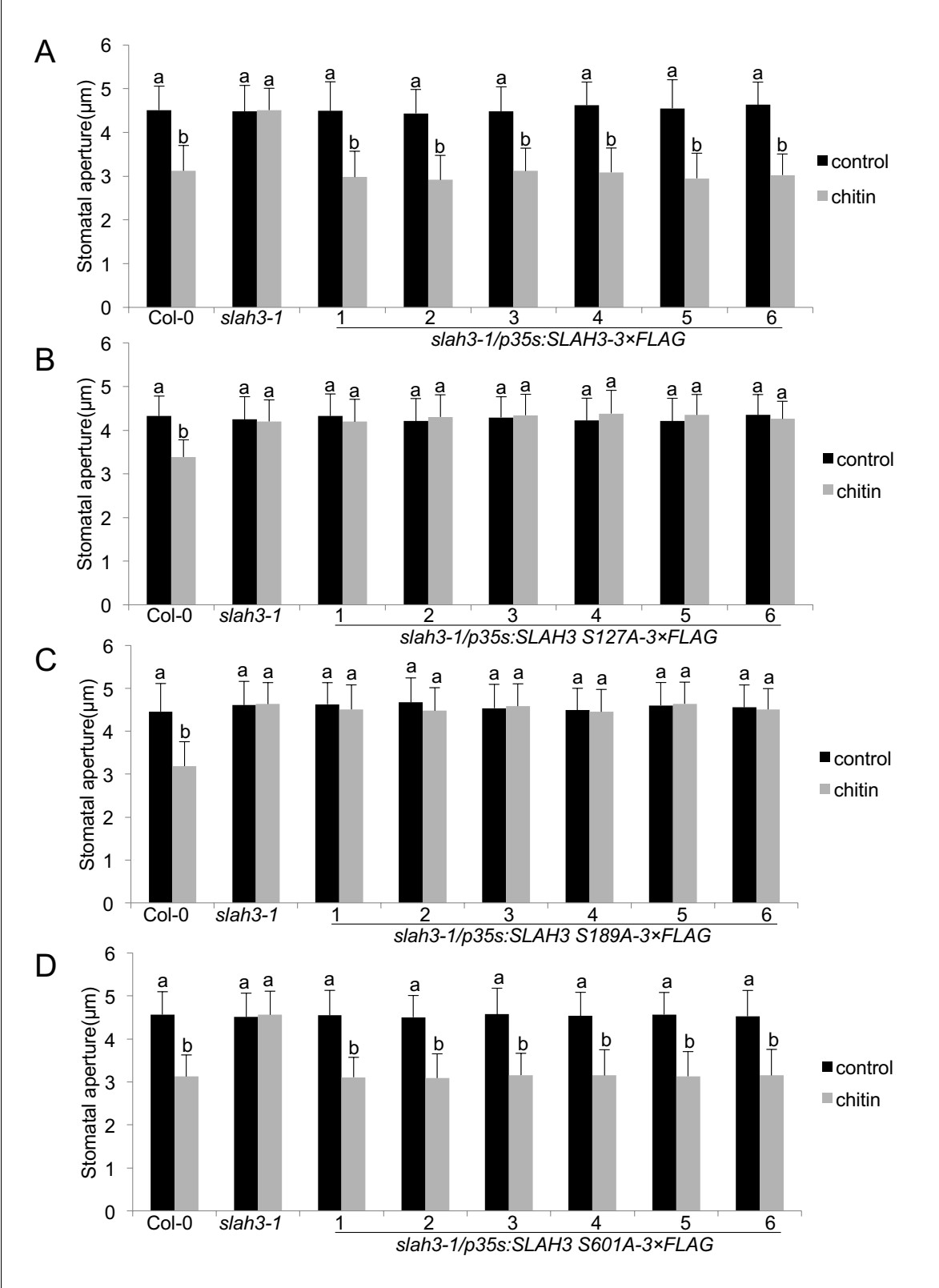

**Figure 4.** SLAH3 phospho-sites S127 and S189 are necessary for chitin-induced stomatal closure. (**A–D**) Stomatal aperture measurements in transgenic *slah3-1/35S::SLAH3−3 × FLAG* wild type (**A**), S127A (**B**), S189A (**C**) and S601A (**D**) variants of SLAH3. Mature leaf discs of six independent transgenic T1 lines were soaked in opening buffer (10 mM MES, 50 mM KCl, pH 6.15) and kept under light (100 μmol.m-2 s −1) for 2 hr. Stomatal aperture was

*Figure 4 continued on next page*

*Figure 4 continued*

measured 2 hr after treatment with 1 mg/ml chitin. Values are mean ± SD (n > 50; two-way ANOVA). Different letters indicate significantly different values at p<0.05. These experiments were repeated three times with similar results.

DOI: https://doi.org/10.7554/eLife.44474.018

The following figure supplement is available for figure 4:

**Figure supplement 1.** Functional complementation of SLAH3 transgenic lines.

DOI: https://doi.org/10.7554/eLife.44474.019

primarily the result of impaired ion fluxes, as in the case of stomatal closure (*Figure 1C*), rather than through altered MAPK signalling (*Meng et al., 2013*). We also found that the chitin-induced ROS burst was not altered in *pbl27* mutants (*Figure 5D*), consistent with previous studies (*Shinya et al., 2014*) and that PBL27 showed no interaction with RBOHD (*Figure 2A*). Although a higher order *rlck vii-1* mutant was partially reduced in ROS production to chitin (*Shinya et al., 2014*; *Rao et al., 2018*), it is the members of the VII-4 subgroup that play major roles in the chitin-induced ROS burst (*Rao et al., 2018*). Thus, the failure of *pbl27* to close stomata in response to chitin is not explained by a compromised ROS burst, supporting a role for SLAH3. Examination of *B. cinerea* infection in *pbl27* revealed no significant differences compared with wild type (*Figure 5E*). This might not be unexpected in light of PBL1 also activating SLAH3 (*Figure 3—figure supplement 1C*). Furthermore, resistance to *B. cinerea* involves PEPR1/2 (*Gravino et al., 2017*), receptors that detect danger signals from the infection and that induce stomatal closure depending on SLAH3 and guard cell-specific SLAC1 (*Zheng et al., 2018*). We would argue that PBL27 mediates activation of SLAH3 in response to CERK1 signalling while PBL1 activates SLAH3 upon PEPR1/2 signalling, suggesting specific signalling pathways that converge on SLAH3. In agreement with this hypothesis, *slah3* but not *pbl27* show compromised resistance to *B. cinerea* (*Figure 5A and E*). Our infection studies revealed that S127 and S189 are important for SLAH3 function in resistance to *B. cinerea*. Thus, the same phospho-sites regulate chitin-induced stomatal closure and anti-fungal immunity at the whole leaf level. Although indicative, it remains to be tested whether SLAH3-mediated stomatal closure directly contributes to anti-fungal immunity.

## Discussion

Guard cells are highly immunomodulatory, expressing PRR complexes that upon MAMP-triggered signalling generate inward currents by S-type anion channels (*Faulkner and Robatzek, 2012*; *Koers et al., 2011*; *Guzel Deger et al., 2015*). SLAC1 and SLAH3 activity is mostly regulated through phosphorylation at their N-termini, mediated by kinases sensitive and insensitive to elevated cytosolic calcium levels (*Geiger et al., 2010*; *Scherzer et al., 2012*; *Maierhofer et al., 2014a*; *Geiger et al., 2011*; *Maierhofer et al., 2014b*). Here, we show that SLAH3 is activated by PBL27, a member of the RLCK subgroup VII-1 (*Rao et al., 2018*), interacting with and functioning downstream of the LYK5-CERK1 chitin-binding receptors (*Yamada et al., 2016*; *Shinya et al., 2014*). Our data are consistent with a direct activation of SLAH3 by PBL27, which is amplified by signalling-active CERK1. In such a model, chitin binding activates the LYK5-CERK1 receptors, inducing auto-phosphorylation and *trans*-phosphorylation events (*Couto and Zipfel, 2016*), which result in the phosphorylation and activation of PBL27 (*Shinya et al., 2014*). Therefore, only signalling-competent CERK1, but not kinase-dead CERK1, amplified PBL27-mediated SLAH3 opening (*Figure 3C*, *Figure 3—figure supplement 2D*). As PBL27 pre-exists in a complex with SLAH3 independent of chitin stimulus (*Figure 2A,B*), activated PBL27 enhanced the phosphorylation of the SLAH3 N-terminus (*Figure 2D*), which fully opens SLAH3 to release anions (*Figure 3C*). Full activation of SLAH3 appears to be required for chitin-induced stomatal closure and resistance to *B. cinerea* infection (*Figure 3F and G*, *Figure 4*, *Figure 5A*): S127 is a phospho-site required for SLAH3 activation by kinase-active PBL27, and S189 represents a phospho-site, which is sensitive to amplifying SLAH3 opening by CERK1-activated PBL27 (*Figure 3G*). The importance of both phospho-sites is supported by the finding that SLAH3 auto-activation by phospho-mimicry required the combination of S127D and S189D (*Figure 3E*). Since PBL1 can also activate SLAH3 and that *slah3* but not *pbl27* mutants supported *B. cinerea* infection, it is plausible to speculate that S127 and S189 are also involved in PBL1 activation of SLAH3.

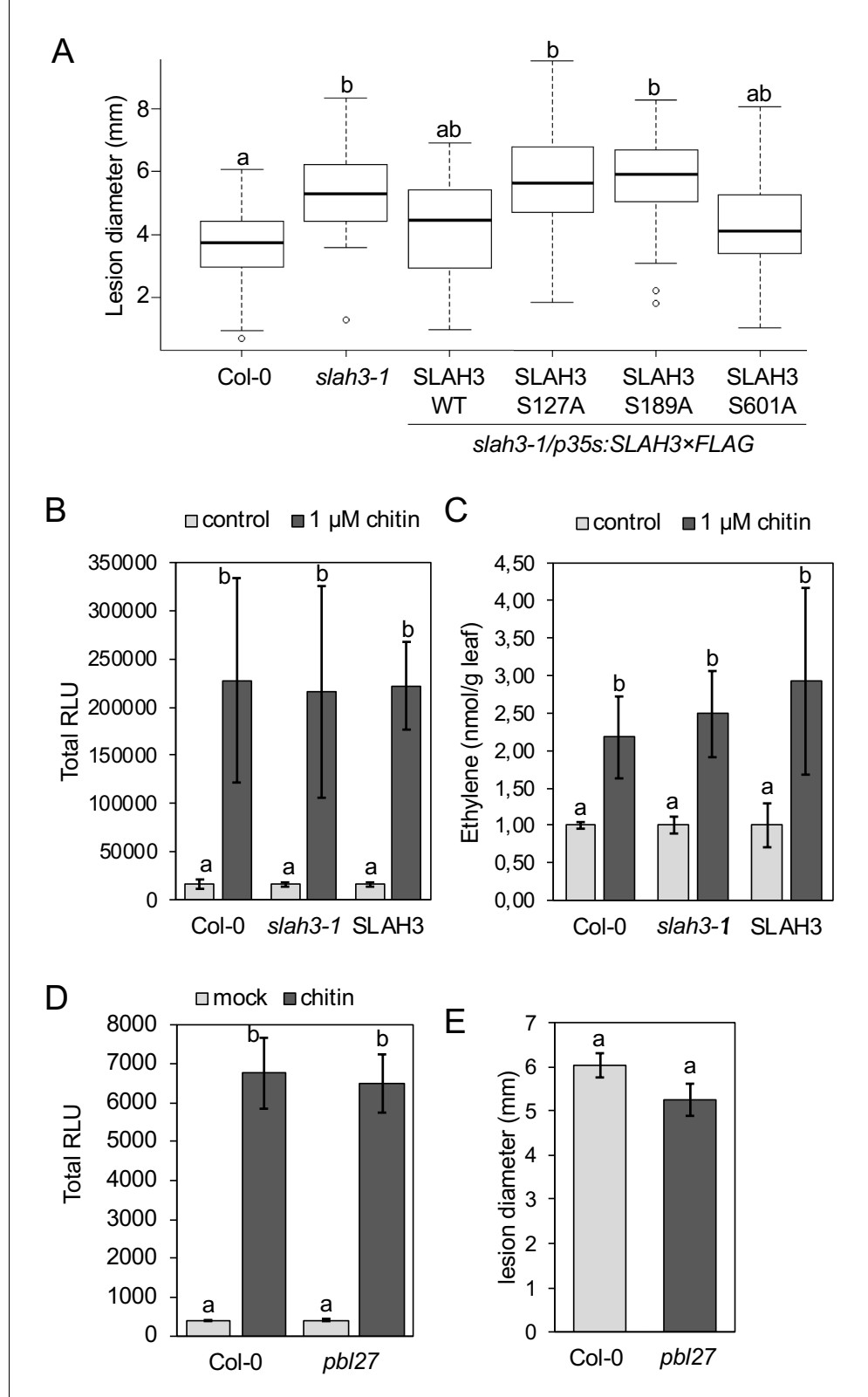

**Figure 5.** SLAH3 mediates resistance to *Botrytis cinerea* in leaves. (**A**) Lesion diameter measurements in *slah3-1* mutants and transgenic *slah3-1/35S::SLAH3–3 × FLAG* wild type (WT), S127A, S189A and S601A variants of SLAH3. Each four mature leaves of 8 to 10 independent transgenic T1 lines were drop-inoculated with *B. cinerea*. Lesion diameter was measured with callipers at 3dpi. Values are mean confidence intervals (n = 4; nested one-way

*Figure 5 continued*

ANOVA). Different letters indicate significantly different values at p<0.05. (B) ROS production measured as relative luminescence units (RLU, integral over 30 min) and (C) Ethylene accumulation in Col-0 WT, *slah3-1* mutants and transgenic *slah3-1/35S::SLAH3−3 × FLAG* wild type (SLAH3) upon chitin treatment. Bars show mean values of 3–5 biological replicates for (B) (n = 4–8) and one for (C) (n = 3). Different letters indicate significantly different values at p<0.05. (D) ROS production measured as RLU in Col-0 and *pbl27-1* mutants upon treatment with 0.1 mg/ml chitin. Bars show data from two biological replicates (n = 2; nine leaf discs per genotype) + /- SEM. (E) Lesion diameter measurements in Col-0 and *pbl27-1* mutants drop-inoculated with *Botrytis cinerea*. Lesion diameter was measured at three dpi. Bars represent average + /- SEM (n = 5 leaves per each genotype). The experiment was repeated twice with similar results. No significant differences (t-test, p<0.05) were detected.
DOI: https://doi.org/10.7554/eLife.44474.020

The following source data is available for figure 5:

**Source data 1.** Source data for Botrytis infection shown in *Figure 5*.
DOI: https://doi.org/10.7554/eLife.44474.021
**Source data 2.** Source data for ethylene measurements shown in *Figure 5*.
DOI: https://doi.org/10.7554/eLife.44474.022
**Source data 3.** Source data for ROS measurements shown in *Figure 5*.
DOI: https://doi.org/10.7554/eLife.44474.023

---

Stomatal closure induced by FLS2 and PEPR1/2 signalling relies on both SLAH3 and SLAC1, and requires BIK1 for activation of anion currents in response to PEPR1/2 signalling (*Guzel Deger et al., 2015*; *Zheng et al., 2018*). However, BIK1 was not sufficient for activation of SLAH3 and SLAC1, and also did not interact with SLAH3 (*Figure 3—figure supplement 1C*, *Figure 2A and B*). OST1 does not activate SLAH3 (*Maierhofer et al., 2014a*; *Geiger et al., 2011*), although it was described to regulate flagellin-induced stomatal closure and confers SLAC1 activation (Figure 3-figure supplement 1B) (*Guzel Deger et al., 2015*; *Geiger et al., 2010*; *Vahisalu et al., 2010*; *Geiger et al., 2009*). Instead, PBL1 belonging to the VII-8 subgroup and closely related with BIK1 activated SLAH3 (*Figure 3—figure supplement 1C*). We may speculate that PBL1 confers SLAH3 activation while OST1, or a closely related SnRK kinase, activates SLAC1. Being directly associated with PRRs (*Yamada et al., 2016*; *Lu et al., 2010*; *Zhang et al., 2010*), it seems that different PRRs integrate different RLCK VII members for distinct signalling outputs (*Rao et al., 2018*), i.e. FLS2-activated BIK1 phosphorylates RBOHD to produce ROS (*Kadota et al., 2014*; *Li et al., 2014*). Consistently, our work identifies a phospho-regulatory pathway that directly links PRR complexes with S-type anion transportation in a specific and signalling-dependent manner. Since CERK1 also functions in LYM1 and LYM3 perception of bacterial peptidoglycan (PGN) (*Willmann et al., 2011*), it will be interesting to explore whether PGN induces stomatal closure through SLAH3 activation by PBL27 or PBL1, thereby promoting anti-bacterial immunity at the level of tissue invasion.

## Materials and methods

### Contact for reagent and resource sharing

Further information and requests for resources and reagents should be directed to and will be fulfilled by the Lead Contacts, Rainer Hedrich (hedrich@botanik.uni-wuerzburg.de) and Silke Robatzek (robatzek@bio.lmu.de).

### Experimental model and subject details

Seeds of wild-type *Arabidopsis* Col-0 and T-DNA insertion mutants of *cerk1* (*cerk1-2*, GABI_096F09), *lyk2* (SALK_152226), *lyk3* (SALK_140374), *lyk4* (WiscDsLox297300_01C), *lyk5* (SALK_ 131911C), *slac1-3* (SALK_099139) and *slah3-1* (GK_371G03) were obtained from the *Arabidopsis* Biological Research Center (ABRC). Seeds of T-DNA insertion mutant *pbl27-1* (GABI_001C07) and *pbl27-1* complementary line *pbl27-1/pPBL27:PBL27−3 × HA* were provided by Dr. Tsutomu Kawasaki (Kindai University, Osaka, JP). To generate *SLAH3* complementary lines, wild type and S217A, S189A and S601A de-phosphorylation *SLAH3* variants were cloned into pW1211 binary vector by LR reaction. The constructs were introduced to *slah3-1* plants through agrobacterium-based transformation

using floral dip method. Plants from six independent T1 *SLAH3* complementation lines were used for RT-PCR analysis and stomatal assays. To generate *CERK1* complementation lines, full length CDS of CERK1 was fused to 500 bp of its native promoter by PCR and cloned to pDnor207 by BP reaction. The DNA fragment was further cloned into pGWB40 binary vector by LR reaction. The construct *pCERK1::CERK1-eGFP* was introduced to *cerk1* plants through agrobacterium-based transformation using floral dip method. The same strategy was used to create *LYK5* and *PBL27* complementary lines. The native promoters of *LYK5* and *PBL27* were 1.1 kb and 1.2 kb. Plants from three independently GFP-expressing T1 lines were used for stomatal assays and subcellular localization.

Investigations on SLAH3 wild type and mutant anion channel function were performed in oocytes of the African clawfrog *Xenopus laevis*. Permission for keeping *Xenopus* exists at the Julius-von-Sachs Institute and is registered at the Regierung of Unterfranken (#70/14). Mature female *Xenopus laevis* frogs were kept at 20°C at a 12/12 h day/night cycle. Mature female *X. laevis* frogs were anesthetized by immersion in water containing 0.1% 3-aminobenzoic acid ethyl ester. Following partial ovariectomy, stage V or VI oocytes were treated with 0.14 mg/ml collagenase I in $Ca^{2+}$-free ND96 buffer (10 mM HEPES pH 7.4, 96 mM NaCl, 2 mM KCl, 1 mM $MgCl_2$,) for 1.5 hr. Subsequently, oocytes were washed with $Ca^{2+}$-free ND96 buffer and kept at 16°C in ND96 solution (10 mM HEPES pH 7.4, 96 mM NaCl, 2 mM KCl, 1 mM $MgCl_2$, 1 mM $CaCl_2$) containing 50 mg/l gentamycin. The complementary DNAs (cDNAs) of SLAH3 mutants and PBL27 WT/mutant were cloned into oocyte expression vectors (based on pGEM vectors), by an advanced uracil-excision-based cloning technique as described by *Nour-Eldin et al. (2006)*. For functional analysis, complementary RNA (cRNA) was prepared with the AmpliCap-Max T7 High Yield Message Maker Kit (Cellscript, Madison, WI, USA). For electrophysiological experiments 10 ng of each cRNA was injected into selected oocytes. Oocytes were incubated for 2 days at 16°C in ND96 solution containing gentamycin.

## Method details
### Plant growth conditions
Gas-sterilized *Arabidopsis thaliana* seeds were plated on 1/2 strength Murashige and Skoog medium with 2% sucrose and 0.8% agar, pH 5.7, stratified for 2 d at 4°C and grown at 22°C under long-day conditions (16 hr light and 8 hr dark). One week later, seedlings were transferred to soil and grown at 22° under a 12 hr light/12 hr dark photoperiod. The light intensity is 100 µmol $m^{-2}$ $s^{-1}$.

For kanamycin selection of T1 complementation lines, gas-sterilized seeds were plated on 1/5 strength B5 medium with 0.8% agar, without sucrose, pH5.7, stratified for 2 d at 4°C and grown at a constant temperature of 22°C with long-day conditions (16 hr light and 8 hr dark). One week later, positive seedlings having green cotyledons were transferred to soil and grown under the condition of 12 hr light and 12 hr dark. The light intensity is 100 µmol $m^{-2}$ $s^{-1}$.

For hygromycin selection of T1 complementation lines, gas-sterilized seeds were plated on 1/2 strength Murashige and Skoog medium with 2% sucrose, pH5.7. Plates were covered with foil and stratified for 2 d at 4°C and grown at a constant temperature of 22°C for 4 d. Positive seedlings having long hypocotyls were transferred to soil and grown under the condition of 12 hr light and 12 hr dark.

### Stomatal assays in leaf discs
four leaf discs were collected from mature leaves of 5 weeks old seedlings with 4 mm biopsy punch and floated in stomatal opening buffer (10 mM MES, 50 mM KCl) with abaxial side for 2 hr at 100 µmol $m^{-2}$ $s^{-1}$ light intensity. Leaf discs were transferred into 1 mg/ml chitin in opening buffer (10 mM MES, 50 mM KCl) or fresh opening buffer (control) for 2 hr and photographed on the Leica DM5500b microscope. At least 50 stomata were recorded per genotype. Stomatal aperture was measured with Image J.

### Pathogen infection
For testing antifungal resistance in *slah3* mutants and *SLAH3* transgenic lines, *B. cinerea* spores ($2.5 \times 10^5$) were drop-inoculated on expanded leaves of 5 weeks old *Arabidopsis* plants, and developing disease lesions were measured with callipers 3 days post inoculation. four leaves per plant were inoculated to provide 4–5 measurements per plant, each containing 8 to 10 selected individuals. For testing antifungal resistance in *pbl27-1* mutants, *B. cinerea* spores ($1.75 \times 10^5$) were prepared in

potato dextrose broth, and a 10 µl drop was placed onto the center of 5 weeks old leaves. five leaves per plant were inoculated. At three dpi, images were taken and lesion diameters were measured using ImageJ.

## Split YFP assay

SLAH3, SLAC1, PBL27, BIK1, CERK1 KD, LYK5 and RBOHD were fused to the N and C termini of YFP to produce SLAH3-YFPn, PBL27-YFPc, SLAC1-YFPn, YFPn-RBOHD, CERK1 KD-YFPc and BIK1-YFPc, respectively. CERK1 KD-YFPc/SLAH3-YFPn, PBL27-YFPc/SLAH3-YFPn, BIK1-YFPc/SLAH3-YFPn, CERK1 KD-YFPc/LYK5-YFPn, BIK1-YFPc/YFPn-RBOHD, and PBL27-YFPc/SLAC1-YFPn were transiently expressed in *N. benthamiana* by agro-infiltration. Leaves were collected 2 d after infiltration. The fluorescence was detected using a Leica TCS SP5 confocal laser scanning microscope.

## Immunoprecipitation and western blot analysis

PBL27$-3 \times$ HA was purified as follows: 4 weeks old *pbl27-1/pPBL27:PBL27$-3 \times$ HA* seedlings were treated with or without 1 mg/ml chitin (Nacosy) for 10 min. 1 g tissue was ground in liquid nitrogen and resuspended in extraction buffer (50 mM Tris-HCl pH 7.5, 150 mM NaCl, 10% glycerol, 0.1% Triton X-100, 0.2% Nonidet P-40, 1% PVPP, 6 mM β-mercaptoethanol, 5 mM TCEP, 1 mM EDTA, 50 µM MG132, 20 mM NaF, 20 mM $Na_3VO_4$, protease inhibitor cocktail Complete Mini tablets). Total exact was cleared by centrifugation and incubated with HA beads at 4℃ for 2 hr. The beads were washed with wash buffer (50 mM Tris-HCl pH 7.5, 150 mM NaCl, 6 mM β-mercaptoethanol, 5 mM TCEP). To obtain PBL27$-3 \times$ HA protein bound to HA beads, elution was done with elution buffer (50 mM Tris-HCl pH 7.5, 150 mM NaCl, 10 µg/ml HA peptide).

Co-immunoprecipitation was processed as follows: The infiltrated parts of *N. benthamiana* leaves were harvested, and 1 g leaf tissue was ground in liquid nitrogen and resuspended in extraction buffer (50 mM Tris-HCl pH 7.5, 150 mM NaCl, 10% glycerol, 0.1% Triton X-100, 0.2% Nonidet P-40, 1% PVPP, 6 mM β-mercaptoethanol, 5 mM TCEP, 1 mM EDTA, 50 µM MG132, 20 mM NaF, 20 mM $Na_3VO_4$, protease inhibitor cocktail Complete Mini tablets). Total exact was cleared by centrifugation and incubated with FLAG beads at 4℃ for 2 hr. The beads were washed with wash buffer (50 mM Tris-HCl pH 7.5, 150 mM NaCl, 6 mM β-mercaptoethanol, 5 mM TCEP), then boiled with 1 × Laemmli Buffer. The elution was separated by SDS-PAGE gel.

For immunoblot analysis, proteins were separated by SDS-PAGE in a 10% acrylamide gel and transferred to PVDF membrane at 25 V for 60 min with semi-dry transfer apparatus. The membrane was blocked with TBST containing 5% skimmed milk powder for 1 hr at room temperature. The membrane was then incubated HRP conjugated antibody in TBST containing 5% skimmed milk for 1.5 hr at room temperature. Then wash with TBST for 3 times. Bands were detected with supersignal west pico plus chemiluminescent substrate. Antibodies and the dilutions used in these experiments were as follows: anti-FLAG (HRP) antibody (1:5000), anti-T7 (HRP) antibody (1:5000).

## Recombinant protein production and purification

His6-SLAH3-N, GST-SLAH3-C-His, His6-MBP-PBL27 and His6-MBP-PBL27 KD were purified as follows: Rosetta *E. coli* cells expressing His tag fusion proteins were incubated in lysis buffer (50 mM Tris-HCl, 500 mM NaCl, 5% glycerol, 20 mM imidazole, pH 8.0). After sonication, the lysate was cleared by centrifugation and incubated with Nickel beads for 2 hr. The beads were washed with wash buffer (50 mM Tris-HCl, 500 mM NaCl, 5% glycerol, 20 mM imidazole, pH 8.0,) to remove unbound proteins. To obtain the recombinant protein bound to Ni beads, elution was done with elution buffer (50 mM Tris-HCl, 500 mM NaCl, 5% glycerol,200mM imidazole, pH 8.0). Finally, eluted proteins were changed to storage buffer (20 mM Tris-HCl, 150 mM NaCl, pH7.5) and concentrated.

## In vitro kinase assay

Purified proteins (2 µg kinase and 2 µg substrate) were diluted in 1x kinase buffer (50 mM Tris pH 7.5, 3 mM $MnCl_2$) up to 20 µl, then added 5 µl 5x kinase buffer (25 mM $MnCl_2$, 5 mM DTT, 5 µM cold ATP and 183 KBq [γ-$^{32}$P] ATP) and incubated 30 min at 30℃ with shaking. Assays were stopped by addition of 5 µl 6x SDS loading buffer and boiling at 70℃ for 10 min. Samples were separated by SDS-PAGE and transferred to PVDF membrane. The signal of [γ-$^{32}$P] ATP was collected with fluorescent image analyzer FujiFILM FLA-5000.

## RT-PCR analysis

To examine the expression of *SLAH3* in *SLAH3* complementary lines by RT-PCR, TURBO DNA-free kit-treated total RNA (5 μg) was denatured and subjected to reverse transcription reaction using SuperScript III (200 units per reaction; Invitrogen) at 50°C for 50 min followed by heat-inactivation of the reverse transcriptase at 70°C for 15 min. PCR amplification was performed using *SLAH3*-specific forward (SLAH3 563 attB1) and reverse primers (Gateway Right FLAG rev) and 28 cycles. Expression levels of *Actin2* were served as an internal control.

## MS analysis

The samples for MS analysis were excised from one dimensional SDS-PAGE gels, stained with colloid Coomassie Brilliant Blue (Simple stain, Invitrogen) and cut to small pieces. They were destained with repeated washing in 50% Acetonitrile. Cysteine residues were modified by 30 min reduction in 10 mM DTT followed by 20 min alkylation with 50 mM chloroacetamide. After extensive washing and dehydration with 50% and 100% Acetonitrile, respectively. The gel slices with modified proteins were incubated with 100 ng of trypsin (Promega) in 50 mM ammonium bicarbonate, 10% Acetonitrile at 37°C overnight. The generated peptides were extracted with 50% Acetonitrile, 5% Formic acid, evaporated to dryness in rotary vacuum evaporator and stored at −20°C.

LC-MS/MS analysis was performed using a hybrid mass spectrometer Orbitrap Fusion (Thermo Scientific) connected to a nanoflow UHPLC system U3000 (Thermo Scientific). Tryptic peptides, dissolved in 2% Acetonitril, 0.2% Trifluoroacetic acid, were injected onto a reverse phase trap column Acclaim Pepmap 100, beads diameter 5 μm, 100 μm x 20 mm (Thermo Scientific) connected to analytical column Acclaim Pepmap 100, beads diameter 3 μm, 75 μm x 500 mm (Thermo Scientific). They were eluted with gradient of 9% to 50% acetonitrile in 0.1% formic acid over 50 min followed by gradient of 50–60% over 3 min at a flow rate of 300 nL*min-1. The mass spectrometer was operated in positive ion mode with nano-electrospray ion source. Molecular ions were generated by applying voltage +2.2kV to a conductive union coupling the column outlet with fussed silica PicoTip emitter, ID 10 μm (New Objective). The ion transfer capillary temperature was set to 275°C and the focusing voltages in the ion optics were in factory default setting.

A method for mass spectrometer has been designed and tested with sensitivity priority for samples of low complexity such as immunoaffinity enriched protein complexes from plants. Therefore, MS events consisted from high resolution full scan in Orbitrap analyser followed by two collisions of 'softer' CID (collision induced dissociation) and more 'energetic' HCD (Higher-energy collisional dissociation) to maximize the chances to acquire spectra with structurally important information. The fragment ions were detected with low resolution detector at the ion trap to achieve maximal speed and sensitivity.

Fusion Software v2.0 was installed. Orbitrap full scan resolution 120000, mass range m/z 300 to 1800 automatic gain control (AGC) for the target 200000 ions and maximal infusion time 50 ms were set. The precursor dissociation events were driven by 'data dependent algorithm' (DDA) with the dynamic exclusion 30 s after the collision had been triggered. The number of precursors selected for collisions were calculated from 3 s duty cycle between full scans, 'Top speed' option, and 'Universal method' settings (AGC = 100, maximal injection time = 500 ms). (Thermo, Poster Note 64608) The isolation width and normalized collision energy for both collision events CID and HCD were set to m/z 1.6 and CE = 30 %respectively. Only precursor ions with positive charge states 2–7 and intensity threshold greater than 10000 were submitted to fragmentation. To improve the quality of phospho-peptide spectra, multi-stage activation (MSA) was used, when neutral loss 98 Da daughter ion was detected in MS2 signal.

## Oocyte assays

### Cloning and site-directed mutagenesis

The complementary DNAs (cDNAs) of SLAH3 WT/mutants, PBL27 WT/mutant, CERK1 WT/mutant, LYK5, BIK1, PBL19, PBL39, or MAPKKK5 were cloned into oocyte expression vectors (based on pGEM vectors), by an advanced uracil-excision-based cloning technique as described by **Nour-Eldin et al. (2006)**. For functional analysis, complementary RNA (cRNA) was prepared with the AmpliCap-Max T7 High Yield Message Maker Kit (Cellscript, Madison, WI, USA). Site-directed mutations were introduced by means of a modified USER fusion method as described by

*Nørholm (2010)* and *Dadacz-Narloch et al. (2011)*. In brief, the coding sequence of the respective anion channel or kinase within an oocyte expression vector (based on pNBIu vectors) was used as a template for USER mutagenesis. Overlapping primer pairs (overlap covering 8 to 14 bp including the mutagenesis site) were designed (*Nour-Eldin et al., 2006*). PCR conditions were essentially as described by *Nørholm (2010)* using PfuX7 polymerase. PCR products were treated with the USER enzyme (New England Biolabs, Ipswich, MA, USA) to remove the uracil residues, generating single-stranded overlapping ends. Following uracil excision, recirculation of the plasmid was performed at 37°C for 30 min followed by 30 min at room temperature, and then constructs were immediately transformed into chemical competent *Escherichia coli* cells (XL1-Blue MRF'). All mutants were verified by sequencing (*Dadacz-Narloch et al., 2011*).

### Double-electrode voltage-clamp (DEVC) measurements
In double-electrode voltage-clamp studies, oocytes were perfused with Mes/Tris-based buffers. The standard solution contained 10 mM Mes/Tris (pH 5.6), 1 mM $Ca(gluconate)_2$, 1 mM $Mg(gluconate)_2$, 1 mM $LaCl_3$ and 100 mM NaCl, $NaNO_3$ or Na(gluconate). To balance the ionic strength, we compensated for changes in the nitrate concentration with Na(gluconate). For recording representative current traces, steady-state currents ($I_{SS}$) and for calculating the voltage dependent relative open probability (rel. $P_O$) standard voltage protocol was as follows: Starting from a holding potential ($V_H$) of 0 mV, single-voltage pulses were applied in 20 mV decrements from +60 to −200 mV. Rel. $P_O$ was calculated from a −120 mV voltage pulse following the test pulses of the standard voltage protocol by fitting the experimental data points with a single Boltzmann equation. The currents were normalized to the saturation value of the calculated Boltzmann distribution. Instantaneous currents ($I_{inst}$) were extracted immediately after the voltage jump from the holding potential of 0 mV to 50 ms test pulses ranging from +70 to −150 mV.

### Immune assays
Ethylene and ROS measurements were conducted as described (*Albert et al., 2010*), with the following modifications: Ethylene accumulation was measured after 5 hr incubation of three leaf pieces per 6 ml tube (three replicates for each condition) containing 500 µl water plus or minus 1 µM chitin resulting in 0,1–0,2 pmol/ml (ethylene in the headspace) for the water controls and 0,3–0,4 pmol/ml after treatment with chitin. For *slah3* mutants, the production of ROS was observed with 20 µM L-012 (Wako) and 2 µg/ml horseradish peroxidase (Applichem) in the reaction assay. Each biological replicate consisted of the mean of at least 4 up to eight single measurements. Chitin (mean degree of polymerization = 7) was prepared as described (*Nars et al., 2013*) and dissolved in water. For *pbl27-1* mutants, ROS production was basically measured as previously described (*Albrecht et al., 2012*). Briefly, ROS was elicited with chitin (0.1 mg/ml) and water treatment was included as a negative control. 6 to 27 leaf discs (4 mm Ø) from 5 weeks old plants were used for each condition, transferred into a 96-well plate and incubated overnight in 200 µl water. The water was then replaced with 100 µl of a luminol/peroxidase solution (17 µg/ml luminol and 10 µg/ml HRP) supplemented or not with chitin (0.1 mg/ml). Luminescence was measured over 60 min using a Tecan Infinite M200 PRO plate reader.

### Current ejection of chitosan
The experiment was carried out with stomata in intact leaves, which were excised from 4 to 5 weeks old plants and fixed in a small petri dish (diameter 35 mm) with double-sided adhesive tape. The leaves were incubated in the bath solution (10 mM KCl, 1 mM $CaCl_2$ and 10 mM MES/BTP, pH 6) and illuminated with 100 µmol $m^{-2}$ $s^{-1}$ white light for at least 2 hr before the start of the experiment. Guard cells in the abaxial epidermis of intact leaves were applied with chitosan by using the current ejection technique (*Huang et al., 2019*), which the tip of a microelectrode was filled with 0.3 mg/mL of chitosan (or 10 mM of MES/BTP, pH6, as control), while the remaining part was filled with 300 mM KCl. The microelectrodes were connected to a custom-made amplifier (Ulliclamp01) via Ag/AgCl half-cells. A reference electrode filled with 300 mM KCl and sealed with 2% agarose (prepared with 300 mM KCl) was bathed to the solution. The microelectrodes were slowly moved towards the cell wall of guard cell under the control of piezo-driven micro-manipulator (MM3A, Kleindiek Nanotechnik). The positively charged chitosan was ejected into the cell wall by application of a current of

1 nA for a period of 60 s. After the ejection, the microelectrode was rapidly removed from the guard cell wall.

## Quantification and statistical analysis

### Plant and oocyte data

All experiments were performed at least three times. Sample size, n, for each experiment was given in the figure legends. Statistical significances for stomatal assays was based on two-way ANOVA post hoc Tukey's HSD test. Statistical significances for pathogenicity assays was based on nested one-way ANOVA test. Statistical significances for oocyte assays was based on one-way ANOVA.

### Software processing and peptide identification

Peak lists in the form of Mascot generic files (mgf files) were prepared from raw data using MS Convert (Proteowizard project) and sent to peptide match search on Mascot server v2.4.1 using Mascot Daemon (Matrix Science Ltd.).

Peak lists were searched against protein databases including typical proteomics contaminants such as keratins, etc. Tryptic peptides with up to two possible miscleavages and charge states +2, +3, +4 were allowed in the search. The following peptide modifications were included in the search: oxidized Methionine (variable), phosphorylated Serine/Threonine/Tyrosine (variable) and carbamido-methylated Cysteine (static). Data were searched with a monoisotopic precursor and fragment ion mass tolerance 10ppm and 0.6 Da respectively. Decoy database was used to validate peptide sequence matches. Mascot results were combined in Scaffold v4.4.0 (Proteome Software Inc) and exported to Excel (Microsoft Office) for further processing and comparisons.

In Scaffold, the peptide and protein identifications were accepted if probability of sequence match and protein inference exceeded 95.0% and 99% respectively. Protein probabilities were calculated in Scaffold by the Protein Prophet algorithm; proteins that contained similar peptides and could not be differentiated based on MS/MS analysis alone were grouped to satisfy the principles of parsimony (*Searle, 2010*).

## Acknowledgements

We like to thank members of the Robatzek and Hedrich laboratories, and Dr. Tsutomu Kawasaki (Kindai University, Osaka, JP) for providing materials. This research was funded by the Gatsby Charitable Foundation (SR), the European Research Council (SR), and the German Research Foundation (DFG) supported grants CRC/TRR166 'ReceptorLight' project B8 (DG, RH), 'Pathogate' (RH), and a Heisenberg Fellowship (SR) by the DFG.

## Additional information

### Funding

| Funder | Grant reference number | Author |
| --- | --- | --- |
| Gatsby Charitable Foundation | Group Leader Fellowship | Silke Robatzek |
| H2020 European Research Council | Project Award | Silke Robatzek |
| Biotechnology and Biological Sciences Research Council | Project Award | Christine Faulkner |
| Deutsche Forschungsgemeinschaft | Project Award | Dietmar Geiger Rainer Hedrich |
| Deutsche Forschungsgemeinschaft | Heisenberg Fellowship | Silke Robatzek |

The funders had no role in study design, data collection and interpretation, or the decision to submit the work for publication.

## Author contributions
Yi Liu, Conceptualization, Formal analysis, Investigation, Methodology, Writing—review and editing; Tobias Maierhofer, Conceptualization, Formal analysis, Investigation, Writing—review and editing; Katarzyna Rybak, Andy Breakspear, Judith Fliegmann, Shouguang Huang, Formal analysis, Investigation; Jan Sklenar, Formal analysis, Investigation, Methodology, Writing—review and editing; Matthew G Johnston, Formal analysis; M Rob G Roelfsema, Supervision, Writing—review and editing; Georg Felix, Conceptualization, Supervision; Christine Faulkner, Conceptualization, Supervision, Funding acquisition, Methodology, Writing—review and editing; Frank LH Menke, Conceptualization, Supervision, Methodology, Writing—review and editing; Dietmar Geiger, Conceptualization, Formal analysis, Supervision, Funding acquisition, Methodology, Writing—review and editing; Rainer Hedrich, Conceptualization, Supervision, Funding acquisition, Project administration, Writing—review and editing; Silke Robatzek, Conceptualization, Formal analysis, Supervision, Funding acquisition, Methodology, Writing—original draft, Project administration, Writing—review and editing

## Author ORCIDs
Matthew G Johnston (iD) http://orcid.org/0000-0003-1141-6135
Christine Faulkner (iD) http://orcid.org/0000-0003-3905-8077
Frank LH Menke (iD) http://orcid.org/0000-0003-2490-4824
Dietmar Geiger (iD) http://orcid.org/0000-0003-0715-5710
Rainer Hedrich (iD) https://orcid.org/0000-0003-3224-1362
Silke Robatzek (iD) https://orcid.org/0000-0002-9788-322X

## Decision letter and Author response
Decision letter https://doi.org/10.7554/eLife.44474.026
Author response https://doi.org/10.7554/eLife.44474.027

## Additional files

### Supplementary files
• Transparent reporting form
DOI: https://doi.org/10.7554/eLife.44474.024

### Data availability
All data generated or analysed during this study are included in the manuscript and supporting files. Source data files have been provided for main and supplemental figures.

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
