## [Decision Letter]

Thank you for submitting your article "Anion channel SLAH3 is a regulatory target of chitin receptor-associated kinase PBL27 in microbial stomatal closure" for consideration by *eLife*. Your article has been reviewed by three peer reviewers, one of whom is a member of our Board of Reviewing Editors, and the evaluation has been overseen by Christian Hardtke as the Senior Editor.

The reviewers have discussed the reviews with one another and the Reviewing Editor has drafted this decision to help you prepare a revised submission.

Summary:

The study provides evidence that the LYK5-CERK1 complex regulates SLAH3 channel activity through PBL27 during chitin signaling. Notably, the phosphorylation of SLAH3 on S127 and S189 appear to play a key role in SLAH3 activation. They also provide evidence that this regulation plays a major role in chitin-induced stomatal closure and disease resistance to *Botrytis cinerea*. There are several issues raised by the reviewers and require attention from the authors:

Essential revisions:

1) A major weakness of the study is with the biological significance of the findings. The proposed role of SLAH3 in stomatal movement and disease resistance to *Botrytis cinerea* are not properly integrated, since this fungus does not utilize stomata for invasion. A role of SLAH3 in other aspects of PTI should be rigorously tested. For example, is the SLAH3 anion channel activity required for chitin-induced MAPK activation, defense gene expression, ROS burst and whether any of these defenses contribute to resistance to *B. cinerea*. To support a role of PBL27-regulated SLAH3 activation in *B. cinerea* resistance, *pbl27* needs to be tested to see whether it is compromised in resistance.

2) It is puzzling that S189A confers constitutive active channel activity, but the transgenic plants display a constitutive open stomata phenotype. Isn't it supposed to be constitutive closed stomata? In addition, constitutive activation of SLAH3 channel by S189A suggests that phosphorylation on S189 negatively regulates channel activity, which works against the idea that PBL27 positively regulates SLAH3. Please clarify.

3) In all BiFC and current experiments, protein levels need to be shown to rule out the possibility that the lack of protein-protein interaction or current is caused by low levels of protein.

4) It is not known whether SLAH3 is phosphorylated in vivo upon chitin treatment. The authors are advised to perform phos-tag or other assays to shown this is indeed the case. If so, it is then necessary to show whether S127 and S189 are major phosphosites in vivo and whether this chitin-induced phosphorylation requires PBL27.

5) In Introduction, Results, and Discussion, the authors cite that PBL27 is a major component in chitin signaling. This has been challenged. A role of PBL27 and the entire clade of RLCK VII-1 in MAPK activation, ROS, and selected gene expression could not be detected. Previous studies have not examined stomatal closure and anion channel activity in the *pbl27* mutant. Also, the sentence "Similarly, chitin stimulates S-type anion channels (Koers et al., 2011), but the molecular components involved in channel activation….remain elusive". This is a key information item, but the cited original study (Koers et al., 2011) used 'chitosan', a deacetylated derivative of chitin, to induce stomata close. Although both are components of fungal cell wall, they are different chemicals and may initiate different signaling steps. It was shown that chitosan may not induce CERK1-dependent immune response (Kaku et al., 2006, and Petutschnig et al., 2010). Authors need to clarify this point to avoid misleading readers. The authors need to be concise about the literature.

---

## [Author Response]

Essential revisions:1) A major weakness of the study is with the biological significance of the findings. The proposed role of SLAH3 in stomatal movement and disease resistance to Botrytis cinerea are not properly integrated, since this fungus does not utilize stomata for invasion. A role of SLAH3 in other aspects of PTI should be rigorously tested. For example, is the SLAH3 anion channel activity required for chitin-induced MAPK activation, defense gene expression, ROS burst and whether any of these defenses contribute to resistance to B. cinerea. To support a role of PBL27-regulated SLAH3 activation in B. cinerea resistance, pbl27 needs to be tested to see whether it is compromised in resistance.

Thank you for your insights and suggestions. Given the limited availability of a fungal pathosystem to address stomatal infecting fungi in Arabidopsis, we utilized resistance to *Botrytis cinerea* in whole leaves to show a role of SLAH3 in immunity. We agree that it is an important point to ask whether SLAH3 generally compromises immune defences in response to chitin. We now provide two lines of evidences that SLAH3 is not compromised in prototypic PTI responses: Firstly, chitin induces a wild type-like ROS burst in *slah3* mutants. This shows that both impaired stomatal closure and resistance to *B. cinerea* is not a result of possible altered ROS production. Secondly, chitin induces wild-type like ethylene production in *slah3*. Because ethylene production is downstream of MAPK activation (Liu and Zhang, 2004), it suggests that impaired stomatal closure and anti-fungal immunity in *slah3* are not primarily affected at the level of MAPK activation. We also tested chitin-induced ROS and resistance to *B. cinerea* in *pbl27*, which were wild type-like. We therefore exclude the possibility that altered ROS production could impair stomatal closure in *pbl27*. We speculate that *pbl27* shows wild type-like resistance to *B. cinerea*, because of functional redundancy with PBL1, since we found that this RLCK also activates SLAH3. We have included this information in new Figure 5 and Figure 3—figure supplement 1C, and have made respective changes in the text.

2) It is puzzling that S189A confers constitutive active channel activity, but the transgenic plants display a constitutive open stomata phenotype. Isn't it supposed to be constitutive closed stomata? In addition, constitutive activation of SLAH3 channel by S189A suggests that phosphorylation on S189 negatively regulates channel activity, which works against the idea that PBL27 positively regulates SLAH3. Please clarify.

We agree that auto-active SLAH3 S189A could be expected to trigger continual stomatal closure.

However, finding that CERK1 amplified PBL27-mediated SLAH3 activation depending on S189A inspired our discussion of PBL27 as a positive regulator. To better describe this, we have modified the text and now state:

“The observation that plants expressing the S189A variant were impaired in stomatal closure is in apparent contrast with its auto-activity (Figure 3F). […] However, although CIPK23/CBL1 were unable to increase SLAH3 opening beyond the level of its autoactivation (Figure 3—figure supplement 3B), we cannot exclude the possibility that the auto-activation of SLAH3-S189A in oocytes hints at a structurally important role in SLAH3 function that is not only specific to the activation of PBL27 by CERK1.”

“In such a model, chitin binding activates the LYK5-CERK1 receptors, inducing autophosphorylation and *trans*-phosphorylation events [Nour-Eldin et al., 2006], which result in the phosphorylation and activation of PBL27 [Shinya et al., 2014]. […] The importance of both phospho-sites is supported by the finding that SLAH3 auto-activation by phosphomimicry required the combination of S127D and S189D (Figure 3E).”

3) In all BiFC and current experiments, protein levels need to be shown to rule out the possibility that the lack of protein-protein interaction or current is caused by low levels of protein.

We agree that it is important to show protein accumulation in order to draw conclusions. For BiFC experiments, all constructs used showed at least in one of the tested interactions a positive signal. This shows that the constructs are generally expressed in the conditions used. Lacking a combination that would show a positive interaction with SLAC1, we replaced PBL27-YFPc + SLAC1-YFPn with PBL27-YFPc + YFPn-RBOHD. This revealed no BiFC signal, which is also interesting in light of PBL27 and a possible role in ROS burst. For current experiments, we used in all cases but one (SLAC1-YFPc and OST1-YFPn to promote interaction) untagged constructs. This was motivated to avoid possible interferences on current generation caused by the tag. Importantly, we directly injected the constructs in the form of cRNA, as *Xenopus oocytes* very efficiently translate cRNAs (Tammaro et al., 2008). This information is included in the text.

4) It is not known whether SLAH3 is phosphorylated in vivo upon chitin treatment. The authors are advised to perform phos-tag or other assays to shown this is indeed the case. If so, it is then necessary to show whether S127 and S189 are major phosphosites in vivo and whether this chitin-induced phosphorylation requires PBL27.

Thank you for your suggestion. We agree that it is important to show whether SLAH3 is phosphorylated in vivo. Using immunoprecipitation from SLAH3-FLAG transgenic plant (Figure 4—figure supplement 1) and Phos-tag SDS, we detected a band shift of SLAH3-FLAG in response to chitin, which was sensitive to phosphatase treatment. This suggests that chitin induces SLAH3 phosphorylation in vivo. We have included this information in new Figure 2E and have made respective changes in the text.

Unfortunately, our attempts to show differences in SLAH3 phosphorylation using

immunoprecipitation of FLAG-tagged SLAH3-S127A and SLAH3-S189A from respective transgenic plants (Figure 4—figure supplement 1) were not successful. We believe that the resolution of Phos-tag SDS is too small to detect such differences at individual residues.

Having no SLAH3-FLAG transgenic line in the *pbl27* mutant background, we attempted transient expression of the tagged channel in protoplasts from *pbl27*. However, expression levels of SLAH3FLAG from transfected protoplasts in both wild type and *pbl27* were very low and not sufficient to reveal a phosphorylated band of SLAH3-FLAG using Phos-tag SDS.

However, having shown that:

- chitin-induces SLAH3 phosphorylation in vivo (Figure 2E),

- PBL27 promoting phosphorylation of the SLAH3 N-ter in a chitin-dependent manner using an in vivo*-*in vitro approach (Figure 2D),

- PBL27 phosphorylating the SLAH3 N-ter at S127, S189 and S601 in vitro (Figure 2C),

- PBL27 mediating SLAH3 activation depending on S127 in a heterologous system (Figure 3F),

- PBL27 kinase dead version does not activate SLAH3 in a heterologous system (Figure 3B),

- PBL27 mediating CERK1-dependent amplification of SLAH3 requiring S189 in a heterologous system (Figure 3G),

- SLAH3-S127A and SLAH3-S189A compromising chitin-induced stomatal closure and resistance to fungal infection in planta (Figures 4B, 4C, 5A),

it would be consistent to conclude that S127 and S189 are major phospho-sites targeted by PBL27 in chitin signalling.

We appreciate that this does not address concerns over the functional redundancy in the VII subfamily (Rao et al., 2018). We therefore explored additional members of the subgroups VII-4 (PBL19 and PBL39) and VII-8 (PBL1 in addition to BIK1) for their potential to activate SLAH3. Only PBL1, which is associated with FLS2 and PEPR1/2 signalling, conferred SLAH3 activation. This suggests that some of the other PBLs with roles in chitin signalling are not involved in SLAH3 activation (Rao et al., 2018). Moreover, it suggest that PBL1 could mediate SLAH3 activation i.e. in the context of stomatal closure to flagellin and danger trigger (Guzel Deger et al., 2015) (Zheng et al., 2018). We have included this information in new Figure 3—figure supplement 1C and have made respective changes in the text.

5) In Introduction, Results, and Discussion, the authors cite that PBL27 is a major component in chitin signaling. This has been challenged. A role of PBL27 and the entire clade of RLCK VII-1 in MAPK activation, ROS, and selected gene expression could not be detected. Previous studies have not examined stomatal closure and anion channel activity in the pbl27 mutant. Also, the sentence "Similarly, chitin stimulates S-type anion channels (Koers et al., 2011), but the molecular components involved in channel activation….remain elusive". This is a key information item, but the cited original study (Koers et al., 2011) used 'chitosan', a deacetylated derivative of chitin, to induce stomata close. Although both are components of fungal cell wall, they are different chemicals and may initiate different signaling steps. It was shown that chitosan may not induce CERK1-dependent immune response (Kaku et al., 2006 and Petutschnig et al., 2010). Authors need to clarify this point to avoid misleading readers. The authors need to be concise about the literature.

Thank you for your suggestion. We consolidated our information on PBL27 and other RLCKs with roles in chitin signalling and immunity throughout the text. To better alert the readership about findings related to chitosan, we made revisions in the text and also included data showing that chitosan induces stomatal closure in Arabidopsis, which was partially depending on SLAH3. We have included this information in new Figure 1—figure supplement 2 and have made respective changes in the text.